# Examining the Role of the Noradrenergic Locus Coeruleus for Predicting Attention and Brain Maintenance in Healthy Old Age and Disease: An MRI Structural Study for the Alzheimer’s Disease Neuroimaging Initiative

**DOI:** 10.3390/cells10071829

**Published:** 2021-07-20

**Authors:** Emanuele R. G. Plini, Erik O’Hanlon, Rory Boyle, Francesca Sibilia, Gaia Rikhye, Joanne Kenney, Robert Whelan, Michael C. Melnychuk, Ian H. Robertson, Paul M. Dockree

**Affiliations:** 1Department of Psychology, Trinity College Institute of Neuroscience, Trinity College Dublin, Llyod Building, 42A Pearse St, 8PVX+GJ Dublin, Ireland; OHANLOER@tcd.ie (E.O.); BOYLER1@tcd.ie (R.B.); rikhyeg@tcd.ie (G.R.); KENNEYJ@tcd.ie (J.K.); melnychm@tcd.ie (M.C.M.); iroberts@tcd.ie (I.H.R.); DOCKREEP@tcd.ie (P.M.D.); 2Department of Psychiatry, Royal College of Surgeons in Ireland, Hospital Rd, Beaumont, 9QRH+4F Dublin, Ireland; 3Department of Psychiatry, School of Medicine Dublin, Trinity College Dublin, 152-160 Pearse St, 8QV3+99 Dublin, Ireland; fsibilia@tcd.ie; 4Department of Psychology, Global Brain Health Institute, Trinity College Dublin, Lloyd Building, 42A Pearse St, 8PVX+GJ Dublin, Ireland; Robert.Whelan@tcd.ie

**Keywords:** locus coeruleus, reserve, brain age, visual attention, Alzheimer’s disease, mild cognitive impairment, normal aging, neuroimaging, voxel based morphometry

## Abstract

The noradrenergic theory of Cognitive Reserve (Robertson, 2013–2014) postulates that the upregulation of the locus coeruleus—noradrenergic system (LC–NA) originating in the brainstem might facilitate cortical networks involved in attention, and protracted activation of this system throughout the lifespan may enhance cognitive stimulation contributing to reserve. To test the above-mentioned theory, a study was conducted on a sample of 686 participants (395 controls, 156 mild cognitive impairment, 135 Alzheimer’s disease) investigating the relationship between LC volume, attentional performance and a biological index of brain maintenance (BrainPAD—an objective measure, which compares an individual’s structural brain health, reflected by their voxel-wise grey matter density, to the state typically expected at that individual’s age). Further analyses were carried out on reserve indices including education and occupational attainment. Volumetric variation across groups was also explored along with gender differences. Control analyses on the serotoninergic (5-HT), dopaminergic (DA) and cholinergic (Ach) systems were contrasted with the noradrenergic (NA) hypothesis. The antithetic relationships were also tested across the neuromodulatory subcortical systems. Results supported by Bayesian modelling showed that LC volume disproportionately predicted higher attentional performance as well as biological brain maintenance across the three groups. These findings lend support to the role of the noradrenergic system as a key mediator underpinning the neuropsychology of reserve, and they suggest that early prevention strategies focused on the noradrenergic system (e.g., cognitive-attentive training, physical exercise, pharmacological and dietary interventions) may yield important clinical benefits to mitigate cognitive impairment with age and disease.

## 1. Introduction

Neurodegeneration does not always affect cognitive impairment and daily life functioning to the same extent [1,2,3]. In fact, some individuals even with marked brain deterioration present relatively preserved cognitive function compared to healthy individuals. The observation that the extent of brain deterioration does not reliably or linearly predict the severity of cognitive dysfunction and symptomatology can be viewed as a “cognition-pathology gap” [4]. This phenomenon has been partly explained by the role of compensatory neural processes, underscored by genetic and/or environmental factors that help mitigate the effects of advancing pathology. This resilience to brain damage has been conceptualised by Yaakov Stern (2002) [5] as a model of reserve that moderates between the degree of neural pathology and clinical outcome. Reserve is conceived as a protective factor shaped by cumulative improvement of neural resources due to genetic and environmental factors over a lifetime. In the earlier stages of the development of this construct, it has been differentiated as cognitive reserve and brain reserve [6]. Cognitive reserve refers broadly to a person’s adaptability (i.e., efficiency, capacity, flexibility of cognitive processes) that helps to explain differential susceptibility of cognitive abilities or day-to-day function to brain aging, pathology or insult [7,8]. Cognitive reserve is often represented by intelligence quotient (I.Q.), educational level, occupational complexity and cognitive performance on tests of attention, memory and executive functioning. By contrast, brain reserve is quantified at the neurobiological level in terms of, for example, brain volume, and the degree of functional and structural connectivity between diverse brain regions. However, the current consensus is moving towards a broader and more comprehensive concept of reserve, since neural and cognitive components are deeply integrated with each other. Accordingly, the current terminology, referring to reserve as the cumulative improvement due to genetic and environmental factors pertains to both neurobiological and cognitive levels of reserve [8]. Two important sub-components of reserve are maintenance and compensation [9]. Maintenance is considered to reflect how well a brain is preserved in structural and functional terms, and compensation is the capacity to recruit existing resources with greater efficiency or to employ alternative neural networks in response to cognitive demand. These two sub-components are therefore measured in different ways [8]. First, an index of brain maintenance has been developed to assess the degree of brain deterioration relative to chronological age (Brain Predicted Age Difference—BrainPAD or Brain Gap Estimation—BrainAGE) in order to address a more precise value of maintenance than just brain volume alone [10,11,12]. Based on the brain deterioration in optimal normal ageing, BrainPAD is a cross-sectional measure, which compares an individual’s structural brain health, reflected by their voxel-wise grey matter density, to the state typically expected at that individual’s age. Therefore, higher discrepancies between the biological brain age and chronological age are indices of abnormal ageing and lower reserve with respect to the brain maintenance sub-component. Second, compensation can be indexed by upregulation of a particular neural network or by alternative patterns of functional connectivity during a task, yielding a beneficial behavioural outcome [8,9]. Thus, it can be inferred that such patterns of activation or re-organisation serve compensatory processes.

The evidence of the last decades has shown that the most common and well addressed predictors of reserve—higher education, I.Q., cognitive stimulation, social interaction and physical activity—are related to reduced risk of dementia, better cognitive functioning and overall greater brain mass [13]. Moreover, these factors mediate the gap between brain pathology and cognitive functioning. In the extensive review by Livingstone and colleagues [13], it was estimated that among the 35% of the modifiable risk factors of dementia, high education has a mitigating impact of 8% on disease severity, which increases resilience. Although progress has been made in identifying the contribution of these genetic and/or life experience variables to the accumulation of reserve, the neurobiological substrate underpinning reserve and the extent to which it can mitigate neurodegeneration are not yet understood.

In this regard, Robertson [1,2] proposed the noradrenergic theory of cognitive reserve, arguing that the continuous upregulation of the noradrenergic system throughout the life-time can be a key component of cognitive reserve. The noradrenergic (NA) system originates in a small bilateral nucleus of the pons of the brain stem named the locus coeruleus (LC). LC synthesizes approximately 70% of noradrenaline (NA) in the brain and is responsible of the 90% of the total brain efflux of NA [14]. NA is one of the most diffusely distributed neurotransmitters in the brain and underpins arousal, alertness and attention [14,15,16,17]. Furthermore, several studies of neurodegeneration have found that NA supply and LC integrity are significantly reduced in Alzheimer’s disease [18,19,20,21]. Tau pathology in the LC has been found earlier than in other brain regions during the preclinical stages of the disease [20,22,23,24,25,26], and reduced volume of the LC has been associated with the extent of beta-amyloid and tau pathologies [27,28,29], suggesting it as possible biomarker of neurodegenerative diseases [30]. In a post-mortem study on 165 individuals, Wilson et al. [31] found that higher neural density in the LC was significantly associated with better baseline cognitive functioning and with slower cognitive decline. There is also evidence that the LC exhibits neural loss up to 80% greater in Alzheimer’s patients than in controls [32]. Comparatively, both neuroimaging and histopathological human and animal studies also report that lower LC volume was associated with increased beta-amyloid and tau pathologies across the brain and with increased overall brain deterioration and inflammation [25,28,29,33,34,35,36,37]. Furthermore, murine models demonstrate that NA promotes phagocytosis of beta-amyloid plaques, suggesting that the LC–NA system may offer protection against the development of these plaques in the human brain [38].

Robertson’s theory links the recent evidence concerning the LC–NA involvement in neurodegeneration with its primary role in attention, learning and memory consolidation. The key cognitive reserve factors (higher IQ and education, life-long social and cognitive stimulation and physical exercise) are components, which require activation of the LC in producing and releasing NA [1,2,13,39]. These reserve proxies require arousal modulation and attentional processing in response to problem solving and novelty. In Robertson’s theoretical framework, the continuous protraction of these activities, on a regular basis throughout life-time, can lead to an overall higher noradrenergic tone, facilitating more active cognition and building more resilient neurobiological networks in the face of age- and disease-related decline. The ability to sustain attention and the arousal in response to cognitive engagement and novelty may increase general awareness and mental stimulation, both of which positively contribute to the construct of reserve. Therefore, the proposed neurobiological mechanism behind the protective action of reserve indices may in part be explained by the properties of a more active noradrenergic system, which optimises structural and functional brain connectivity, providing greater resistance to neurodegeneration. NA has been known for its neuroprotective effects [40,41,42], which reduce brain inflammation and promote neurogenesis and synaptogenesis increasing brain derived neurotrophic factor (BDNF), which could increase the number of brain cells [43,44,45,46,47,48]. This noradrenergic system may therefore be critical in to reduce the brain’s vulnerability to both the normal ageing process and pathological neurodegeneration via a more active attentional system [1,2,49]. Therefore, in this theoretical framework, greater LC volume is proposed to reflect a greater noradrenergic tone; conversely, lower LC volume is associated with lower NA levels compromising cognition and brain heath, according to previous studies [17,18,20,21,22,25,26,28,30,33,36,47,49,50,51,52,53,54,55,56,57,58,59,60,61].

The main aim of this study was to test the hypothesized relationship between the structural integrity of the LC and indices of reserve and attention in healthy older controls (HC) and patients with mild cognitive impairment (MCI) and Alzheimer’s disease (AD). To this end, we adopted a neuroimaging voxel-based morphometry (VBM) approach utilizing 3T T1-weighted structural MRI scans from 686 subjects [*n* = 395 (HC), *n* = 156 (MCI), and 135 (AD)] provided by Alzheimer’s Disease Neuroimaging Initiative—ADNI (ADNI2 and ADNI3 phases) [62,63]. Structural volumetric analyses on 3T T1-weighted MRIs with this methodology have been already carried out by several studies showing accurate reliability investigating the integrity of the Brainstem [64,65,66,67], the LC also in the ADNI [68,69,70,71] and the other neuromodulators’ seeds such as the Raphe Nuclei [72,73,74,75], the ventral tegmental area (VTA) [76,77,78,79] and the nucleus basalis of Meynert (NBM) [80,81,82,83,84,85,86,87]. The main objectives of this study were threefold. First (1st branch of VBM analyses), to examine the relationship between LC integrity and cognitive function measured by the Trail Making Test A (TMT-A), which is a clinical tool sensitive to basic attentional efficiency, visual search and motor processing speed [88,89,90]. TMT has been chosen over other attentional tests available in the ADNI since visuospatial engagement of attention more is markedly underpinned by LC–NA system [1,2,91] compared to tasks involving auditory or verbal processing. Furthermore, the LC–NA system expressly innervates the medial prefrontal regions of cortex [14,91,92], which are commonly activated during the spatial attention and visuo-motor requirements of TMT performance in fMRI studies [93,94,95,96]. More specifically, TMT part A has been chosen over the part B since the latter was not completed by all the selected participants included in the ADNI (specifically in the Alzheimer’s group), and also because it is considered to evaluate more structured cognitive processes such as cognitive flexibility and divided attention [97,98]. We anticipated that greater LC volume would be negatively associated with TMT-A completion time in seconds, according to previous findings [31,49,68] and consistent with Robertson’s theory [1,2].

Second (2nd branch of VBM analyses), to examine the extent to which LC integrity mediates the relationship between BrainPAD and attentional efficiency. We tested the hypothesis that higher LC volume would be associated with both higher brain maintenance and better cognitive performance as postulated by Robertson [1,2]. It was expected that higher LC volume would predict more negative values of BrainPAD reflecting ‘younger’ brain age. These analyses investigated the potential compensatory role of the LC–NA system in mediating the relationship between brain maintenance and cognitive performance. Third (3rd and 4th branches of VBM analyses), to analyse the potential relationships between LC integrity and putative indices of reserve, namely, educational level and occupational status held by people throughout their lifetime. It was anticipated that greater LC volume would be associated with both higher levels of education and occupational complexity demands. Indeed, in Robertson’s theory, jobs with higher cognitive demands require greater stimulation of the noradrenergic system contributing to the reserve component [1,2].

As a control procedure, each of the above analyses were repeated testing the opposite relationships as hypothesized, namely, lower LC volume related to better cognition and brain health [88,99,100]. Furthermore, the analyses were repeated for brainstem nuclei of the serotoninergic, dopaminergic and cholinergic systems in order to compare the Noradrenergic hypothesis to the other main neuromodulators involved in cognition. Finally, the analyses were also performed on an ROI drawn in the ventro-rostral portion of the Pons, which to the best of our knowledge is without anatomical nuclei projecting neuromodulators to the cortex, and can therefore be considered a “neuromodulator-free” control region.

Secondary objectives of the study were to investigate which of the above-mentioned neuromodulators showed the greatest covariance with cognitive decline across the three groups (5th branch of VBM analyses) and to explore the differential volumetric variation of the neuromodulators due to gender at different stages of cognitive decline (6th branch of VBM analyses).

## 2. Materials and Methods

Data used in the preparation of this article were obtained from the Alzheimer’s Disease Neuroimaging Initiative (ADNI) database (adni.loni.usc.edu accessed on 15 January 2021). The ADNI was launched in 2003 as a public–private partnership, led by Principal Investigator Michael W. Weiner, MD. The primary goal of ADNI has been to test whether serial magnetic resonance imaging (MRI), positron emission tomography (PET), other biological markers and clinical and neuropsychological assessment can be combined to measure the progression of mild cognitive impairment (MCI) and early Alzheimer’s disease (AD).

### 2.1. Neuroimaging

3Tesla T1-weighted images in Nifti format of ADNI 2 and ADNI 3 phases [62,63] were downloaded from IDA (image data archive powered by LONI—https://ida.loni.usc.edu/ accessed on 15 December 2018). Baseline T1-images of all subjects were selected and organised according to diagnosis: cognitive normal/healthy controls (CN/HC), mild cognitive impairment (MCI) and Alzheimer’s disease (AD). Diagnostic criteria of the ADNI are described by Petersen et al. 2010 [101]. A rigorous manual quality control of the images was carried out according to the rating scale guidelines (1 = poor, 2 = fair, 3 = good, 4 = excellent) of the Human Connectome Project (HCP) (https://www.humanconnectome.org/ accessed on 15 December 2018). Subjects with low definition (excessive blurriness) and/or marked ringing, inhomogeneities and motion artefacts were removed from the dataset. The selected scans were then segmented using CAT12 (Computational Anatomy Toolbox—http://www.neuro.uni-jena.de/cat/ accessed on 15 December 2018) implemented in SPM12. The segmentation was performed using the default CAT12 settings, with a 1 mm isotropic voxel size. The segmentation was run separately for the three groups with the same pipeline and parameters. Subjects with an image quality score below 70% (no more than 0.6% of the sample) were then discarded according to CAT12 reports of quality assurance rating scale (100–90 = A [Excellent]; 90–80 = B [Good]; 80–70 = C [Satisfactory]; 70–60 = D [Sufficient]; 60–50 = E [Critical]; 50–40 = F [unacceptable/failed]. Only the 6.2% of the selected images were below 80% (B). Subsequently, the processed normalized images Grey Matter (GM) + White Matter (WM) were smoothed with a 2 mm^3^ FWHM kernel. Lastly, in order to better account for individual volumetric variability, the Total Intracranial Volume (TIV) was calculated for each subject using CAT12 interface. More details are provided in the Appendix A.

#### 2.1.1. Region of Interest (ROI) Masks

All binary ROIs had a 1 mm^3^ isotropic voxel size and were oriented in the Montreal Neurological Institute (MNI) space as the processed T1 (GM + WM) images. The six ROI masks were obtained by previously published atlases. The technical details and theoretical justifications for the specific ROI definitions are described in the following section for each neuromodulator seed. Further details are provided in the Appendix A.

Accurate MRI localization of the LC in the human brain is still lacking wide-spread agreement [34]. In the last few years, several probabilistic maps of the LC have been released; however, these probabilistic maps are inconsistent in both localization and volume extent within the MNI space. Indeed, different sample sizes have been recruited and this exacerbates the limitations due to different methodologies involved. These differences reflect a large anatomical variability of the samples scanned, suggesting that the LC varies across the general population.

In order to perform volumetric analyses appropriate to the present research and to attempt to resolve these differences, it was necessary to define a common space that included all the previous maps as to increase the likelihood of inclusion of the entirety of the LC, given the probable increase in between-subject anatomical variability in the present ADNI populations (n = 395 HC, n = 156 MCI, n = 135 AD).

Therefore, a new symmetrical “omni-comprehensive” LC mask in the MNI space was created in order to include the whole LC rostro-caudal extent (see Figure 1). Indeed, it was observed that with increasing age, the LC signal intensity tends to shift from the rostral to the caudal portion [34,102]. This process might be influenced by manifold variables, such as ageing, the degree of biological brain maintenance and even dementia progression, which is likely to exacerbate this “caudal-shifting” process. Moreover, it is acknowledged how the noradrenergic system is susceptible to compensatory changes across the brain involving the caudal portion of the LC and peri-coeruleus/LC-peri-dendritic regions (Epi-coeruleus and Sub-coeruleus) [14,54,55,61,103,104,105,106]. Therefore, a larger area rather than a very specific and concise region would be more informative and appropriate while investigating the LC–NA system on different groups, particularly known the heterogeneity of Alzheimer’s disease. The new “omni-comprehensive” LC mask included the six LC MNI atlases previously published: (1) Keren et al. (2009) [102,107], (2) Tona et al. (2017) [108], (3) Betts et al. (2017) [109], (4) Dahl et al. (2019) [33], (5) Liu et al. (2019) [110] and (6) Rong Ye et al. (2020) [111] without encroaching the median raphe (MR) and the dorsal raphe (DR) defined by Beliveau et al. (2015) [112] and the cerebellar white matter. Additionally, the new created LC “omni-comprehensive” mask included the LC meta-mask developed by Dahl et al. (2021) [70], but with a larger and symmetrical rostro-caudal extent to avoid induced lateralization biases in the analyses. Indeed, as pointed out by Betts et al. [109], the LC asymmetries reported in certain MRI studies could be caused by MRI biases of how radiofrequencies are transmitted and received in the scanner. In fact, post-mortem histological studies consistently revealed symmetrical distribution of LC cells [14,113,114,115,116,117].

MR and DR 1 mm^3^ MNI masks were generously provided by Beliveau et al. (2015) [112]. The probabilistic maps and masks were obtained by analysing 232 PET scans matched with high-res 3T structural MRI of healthy subjects between 18 and 45 years old.

The VTA mask was obtained by downloading the VTA MNI probabilistic map from the atlas made by Pauli et al. 2018 [118] from the NeuroVault website (https://neurovault.org/ accessed on 15 December 2018). The atlas was made using the MRI data from the Human Connectome Project (HCP) and was derived from a selected sample size of 168 healthy subjects between 22 and 35 years old. The localisation of the substantia innominata (SI)/NBM was more controversial than the previous nuclei, as there are no specific maps available in MNI space. Albeit, probabilistic MNI maps of the acetylcholine cells of the Forebrain are provided by SPM Anatomy Toolbox 2.2c (https://www.fzjuelich.de/inm/inm1/EN/Forschung/_docs/SPMAnatomyToolbox/SPMAnatomyToolbox_node.html accessed on 15 December 2018) [119]. However, the probabilistic map referring to the SI/NBM (4ch.nii) defined by Zaborszky et al. (2008) [119] overlaps several subcortical nuclei delineated in other atlases [118,120,121,122,123,124,125,126]. Therefore, the “4ch.nii” was used as main reference to delineate the SI/NBM, but was adjusted by excluding the subcortical nuclei identified by other atlases while accounting for the probabilistic localisation of the SI/NBM delineated in previous works [81,82,127,128,129]. In order to control for a “neuromodulator-free” brainstem’s region, a squared binary ROI not referring to any anatomical nuclei was drawn in the ventro-rostral portion of the pons. A greater number of voxels were used in order to obtain a control region similar in voxel-size well-suited for the detection of false positives. The six ROIs are displayed in Figure 2 below.

#### 2.1.2. BrainPAD Measure Calculation

Brain Predicted Age Difference (BrainPAD) is a measure of how the brain is ageing, and it is obtained by calculating the discrepancy between the chronological age and the biological age of the brain defined on healthy brain ageing of typical people. Subjects with a younger brain than their chronological age have negative values, whereas if a subject is ageing faster than their chronological age the index is a positive value. BrainPAD is thought to reflect how well Grey Matter (GM) is maintained, hence it is proposed to be an index of brain maintenance. BrainPAD measure by Boyle et al. (2020) [12] was developed using several datasets. In the first instance they defined the normal GM ageing in healthy subjects. They then trained an algorithm to predict successfully the degree of GM deterioration in relation to the chronological age in further three populations of healthy subjects. The algorithm used in this study is described in detail in Boyle et al. (2020) [12].

#### 2.1.3. Voxel Based Morphometry (VBM) Analyses

VBM analyses were performed using T1 whole brain images (WM+GM), after being pre-processed and smoothed with a 2 mm^3^ FWHM kernel. Each set of analysis aimed to investigate five main research interests, based on Robertson’s theoretical framework [1,2]. Each analysis first considered the LC and was then repeated separately for the other brainstem nuclei and the CP ROI as a control procedure testing both positive and negative relations. The statistical thresholds were settled at *p* < 0.01, and later increased progressively until the results disappeared (namely: *p* < 0.001, *p* < 0.01 *FWE*, *p* < 0.001 *FWE*).

#### 2.1.4. Relationship between LC Volume and Attention

The first question addressed by this study was to investigate whether the LC volume can be a predictor of attentional performances measured with TMT-A (visuo-motor speed processing). Three multiple regression models were run for each group. In the model, TMT-A was included as a continuous variable and TIV, education and age were entered as covariates. Then, based on previous literature and the main hypothesis, a negative relation between the LC volume and the TMT-A was also investigated, namely, a larger LC volume was expected to be related to faster attentional performances (fewer seconds spent in completing the task). The following contrast was used for negative relation: 0 0 0 0 −1. The contrast: 0 0 0 0 1 was used as well as control analyses testing the positive relation. A further step in the analyses was to indicate the LC mask as “inclusive mask” in order to isolate the LC involvement in the model. Similarly, all the other ROIs were tested in the same way.

#### 2.1.5. Relationship between LC Volume and Biological Brain Maintenance

The negative relation between LC volume and BrainPAD was then tested across the three groups, with the hypothesis that greater LC volume would be associated with lower and negative BrainPAD scores, reflecting reduced brain aging relative to chronological age. As in the previous analyses, BrainPAD was treated as dependent continuous variable and the same covariates and contrasts were used, including chronological age as suggested by Le TT et al. (2018) [130]. The following contrast was used for negative relation: 0 0 0 0 −1. The contrast: 0 0 0 0 1 was used as well as control analyses testing the positive relation. A further step in the analyses was to indicate the LC mask as inclusive in order to isolate the LC involvement in the model. Similarly, all the other ROIs were tested in the same way.

### 2.2. Mediation Analyses

Mediation analyses with multiple parallel mediators were carried out in order to better clarify the possible role of the LC as key component in mediating reserve indices and attentional performances. The analyses considered the three groups separately and were performed using the toolbox PROCESS v3.4 and SPSS macro developed by Andrew F. Hayes (http://www.processmacro.org/ accessed on 15 December 2018) implemented in SPSS 25 (https://www.ibm.com/products/spss-statistics accessed on 15 December 2018). The toolbox PROCESS enables one to perform manifold types of mediation and moderation analyses. In SPSS, from the PROCESS interface it was selected the model number four with 95% confidence intervals and 5000 bootstrap samples. This model number can be used to perform mediation analyses with parallel multiple mediators and covariates. TMT- A time in seconds was used as the Y variable and BrainPAD scores as the X variable. The extracted average volume values of the six ROIs were considered as six parallel mediators. TIV and age were treated as covariates.

### 2.3. Bayes Factors Calculation

In JASP (https://jasp-stats.org/ accessed on 15 January 2019), Bayes Factors (BF) were calculated in order to better discriminate the differential involvement of the six ROIs in the two main domains investigated, visuo-motor speed processing (TMT-A) and biological brain maintenance (BrainPAD). The extracted average volumes of the six ROIs were tested in a Bayesian correlation model in order to establish the strength of the relationships between TMT-A and BrainPAD across the three groups. Further details are provided in the Appendix A.

### 2.4. Rationale for the Neuromodulatory Subcortical System ROI Selection

From dopamine, noradrenaline is synthetized subcortically in the brainstem, specifically in the LC, in the dorsal pontine tegmentum and in the lateral tegmental neurons [14]. However, LC is the main structure responsible for the NA production and also accounts for 90% of its cortical NA innervation. LC’s projections are vastly spread throughout the cortex and the cerebellum [15,91]. For these reasons, it has been defined as the main core structure of investigation of the noradrenergic theory of cognitive reserve. Given that the analyses were designed to be performed also on clinical samples, control analyses were made to account for the possibility to detect false positive concerning the LC involvement in attention and reserve due the ongoing diffuse neurodegeneration of the samples. Therefore, other main neuromodulators and their main core nuclei were considered control regions to better assess the implication the LC–NA system in attention and reserve. The first control neuromodulator defined was the serotoninergic system because it is believed to broadly modulate markedly different processes than NA. Additionally, Serotonin is synthesised in the pons very closely to the LC but from a different precursor (tryptophan) [14]; thus, this anatomical area is particularly well suited for controlling purposes. The median and the dorsal raphe nuclei are the largest serotoninergic seeds producing and projecting serotonin to the cortex and the cerebellum via the basal forebrain [14,112]. The dopaminergic system was also examined, and so the ventral tegmental area was taken as a core area for the analyses. VTA is the main brain nucleus together with the substantia nigra (SN), where dopamine is synthesised from the amino acid tyrosine [14]. The VTA is responsible for the cortical irroration of dopamine while the SN projects subcortically. For this reason, VTA was defined as control region to control for the dopaminergic system. Regarding acetylcholine, the substantia innominata/nucleus basalis of Meynert (NBM) was chosen over the tegmental cholinergic neurons because it has the largest number of cholinergic neurons and it projects diffusely to the entire cortex. More than 90% of the NBM includes cholinergic magnocellular neurons [14,127]. Finally, in order to control for a “neuromodulator-free” brainstem region, an ROI not referring to any anatomical nuclei was designed in the ventro-rostral portion of the pons.

## 3. Results

Descriptive statistics for key demographic, neural and neuropsychological variables are presented in Table 1.

### 3.1. Sociodemographic Characteristics and Indices of Reserve and Attention

Age significantly differed across groups, F (2.683) = 10.13, *p* < 0.001. Post hoc comparisons showed that the HC group was significantly younger than the MCI group (*p* = 0.014) and the AD group (*p* < 0.001). The MCI and AD groups did not significantly differ from each other in age (*p* = 0.68). Groups also differed in education, F (2.683) = 19.41, *p* < 0.001. The average years of education was higher for the HC group compared to the MCIs (*p* < 0.001) and the AD group (*p* < 0.001). The MCI and AD groups did not differ in mean years of education (*p* = 0.68). There was an effect of occupation, F (2.683) = 11.71, *p* = 0.025. The HC showed a higher occupational rank compared to MCI (*p* = 0.031). No other group comparisons for occupation were significant (all *p* > 0.3). A Chi-squared test showed that gender was not evenly distributed across the three groups (Χ^2^ = 12.35; df: 2; *p* = 0.002). As reported in Table 1, the HC contained significantly more females (57%) while in MCI and AD males were more highly represented (57.1% and 54.8%, respectively). There was no significant difference in TIV by group, F(2.683) = 2.62, *p* = 0.07. However, there was a significant difference in BrainPAD scores, F(2.683) = 101.2, *p* < 0.001. There was a systematic pattern in which AD patients showed greater BrainPAD scores (indicating an older brain relative to chronological age) compared to MCI patients (*p* < 0.0001), who in turn showed a greater mean BrainPAD score than the HC group (*p* < 0.0001). All groups significantly differed in time taken to complete the TMT- A, F (2.683) = 127, *p* < 0.001. AD patients exhibited the longest duration (mean seconds) compared to the MCI (*p* < 0.001), which showed a longer mean time-to-completion than the HC group (*p* < 0.001). More details are provided in the Appendix A including the average volumetric and gender differences across the three groups for the six ROIs (reported in Appendix A).

### 3.2. 1st Branch of VBM Analyses: Multiple Regressions—TMT-A (Attention—Visuo-Motor Speed Processing)

Does the LC predict attention performance relative to other neuromodulator seed regions?

As can be observed in Table 2, for the statistical threshold of *p* < 0.001 only the three voxels of the NBM were significant in the HC. For the MCI group, the reduced volume of the LC (142 voxels *p* < 0.01 and 21 voxels for *p* < 0.001) was associated with longer TMT-A completion time. Similarly, in the AD, the strongest predictor of attention performance was the LC with nine (122 voxels for *p* < 0.01) voxels negatively associated with the attentional performance. As shown in Figure 3 the average LC results are localised within a region overlapping the LC core defined by previous atlases. Further neuromodulators’ seeds also contributing to the variance in TMT-A performance included 14 VTA voxels and 19 NBM voxels negatively associated with performance. For the three groups, these results did not hold when FWE correction was applied. As a control measure, the opposite relationships (positive associations with TMT-A) across the three groups for all six ROIs were assessed and revealed no significant results. (More details are provided in the Appendix A).

### 3.3. 2nd Branch of VBM Analyses: Multiple Regressions—BrainPAD (Reserve—Brain Maintenance)

Does the LC predict brain maintenance relative to the other neuromodulator seed regions?

Table 3 shows a significant (*p* < 0.001 threshold FWE corrected) cluster of 153 LC voxels predicting BrainPAD score in the HC group, demonstrating that greater LC volume is associated with a lower or negative (i.e., younger) BrainPAD score. Similarly, 71 voxels of the DR were significant as well. MCI showed a similar trend, with statistical threshold of *p* < 0.001, 59 LC voxels and 109 DR voxels predicting BrainPAD score.

As shown in Figure 4, the average LC results are localised within a region overlapping the LC core defined by previous atlases. A lesser contribution of 2 VTA voxels and 12 NBM voxels also predicted BrainPAD score. In the AD, the most widespread effects were observed. All the ROIs, except the control pontine region, were found to negatively predict BrainPAD scores. At the *p* < 0.001 threshold, the most significant cluster was found in the LC (192 voxels). Two clusters of 90 and 20 voxels were observed in the DR and MR, respectively.

There were also two bilateral clusters in the VTA (nine voxels) and a cluster in the NBM (five voxels). When FWE correction was applied, no results survived for MCI and AD. When tested, the opposite relationships (positive associations with BrainPAD) across the three groups for all six ROIs showed no significant results. (more details are provided in the Appendix A).

### 3.4. Mediation Analyses

Does the LC mediate the relationship between BrainPAD (X) and attention performance (Y)?

A multiple parallel mediation analysis was conducted for each of the three groups. Bootstrap confidence intervals were used to examine the role of the six subcortical nuclei in mediating the relationship between the BrainPAD score and attention performance, while controlling for age and TIV (see Figure 5—schematic mediation pathways for MCI group). In HC and AD, the total effect of X on Y was significant. The direct effects of X on Y were found not significant for the HC and significant for AD groups, indicating that better brain maintenance relative to chronological age was predictive of attention performance (HC total effect of X on Y: 0.1586; se: 0.0552; T:2.874, *p* = 0.0043; LLCI: 0.0501; ULCI: 0.2672; c_ps: 0.0179; c_cs: 0.1378) (HC direct effect of X on Y: 0.1225; se: 0.0877; T:1.3979, *p* = 1.630; LLCI: −0.0498; ULCI: 0.2949; c_ps: 0.0138; c_cs: 0.1065). (AD total effect of X on Y: 1.7088; se: 0.3918; T:4.3608, *p* < 0.0000; LLCI: 0.9336; ULCI: 2.4840; c_ps: 0.0467; c_cs: 0.4262) (AD direct effect of X on Y: 1.2410; se: 0.4658; T:2.6644, *p* = 0.0088; LLCI: 0.3189; ULCI: 2.1630; c_ps: 0.0339; c_cs: 0.3095). However, no indirect effects of the six mediators were apparent. On the other hand, in MCI, the LC alone was found to significantly mediate the relationship between BrainPAD (Y) and TMT-A (X) (indirect effect of X on Y: 0.0927; BootSE: 0.0499; BootLLCI: 0.0111; BootULCI: 0.2043). The total effect of X on Y was also significant (effect: 0.4224; se: 0.1583; t: 2.6678; p: 0.0085; LLCI: 0.1096; ULCI: 0.7352; c_ps: 0.0237; c_cs: 0.2130). Controlling for the mediation effect, the direct effect of X on Y was not significant (direct effect: 0.0383; se: 0.2496; t: 0.1535; p: 0.8782, LLCI: -0.4552; ULCI: 0.5318, c’_ps: 0.0021, c’_cs:0.0193), implying that the effect of BrainPAD on attention performance in MCI is mediated indirectly through the LC volume. This finding suggests that the way brain maintenance affects attention performance in MCI patients is disproportionately influenced by the noradrenergic system compared to other neuromodulatory systems.

**Figure 5 cells-10-01829-f005:**
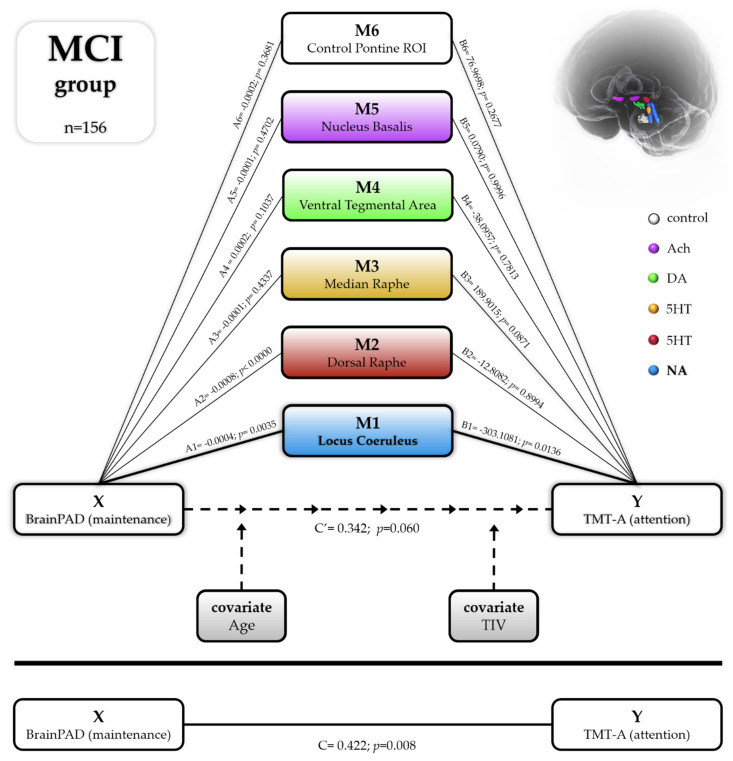
Statistical diagram of the parallel mediation model (model n.4) testing the relation between BrainPAD (biological measure of brain maintenance) and TMT-A (attentional visuo-motor speed processing) in MCI group (n = 156). The average volumes of the six ROIs were treated as parallel mediators in order to test the noradrenergic theory of cognitive reserve versus the other main neuromodulator’s seeds: serotoninergic (Dorsal and Median Raphe), dopaminergic (Ventral Tegmental Area), cholinergic (Nucleus Basalis of Meynert) and additionally with a brainstem control region neuromodulator-free (control pontine ROI). In the MCI group, the only significant mediator of the relation between brain maintenance and cognitive performance was the volume of the LC (Noradrenaline); all the other five ROIs were not significant in mediating the relationship. The model was covaried for age and total intracranial volume (TIV).

**Table 2 cells-10-01829-t002:** VBM multivariate linear regression analyses for the six ROIs across the three groups (HC n = 395; MCI n = 156; AD n = 135). The results were adjusted for total intracranial volume (TIV), and age and education were entered as continuous variables. TMT-A time in seconds was treated as a continuous dependent variable and the structural scans as the independent variable. For a statistical threshold of *p* < 0.001, the table reports the number of significant voxels negatively related to TMT-A time in seconds.

Brain Regions	Side	MNI Coordinates			Peak T Value ^a^	Peak Z-Score ^b^	Peak Cluster Ke ^c^	*p*-Value Uncorr ^d^	FEW ^e^	FDR ^f^	Total Number of Voxels *p* < 0.001 Threshold ^g^
		*x*	*y*	*Z*							
**HC** (n. 395)											
Locus Coeruleus	/	/	/	/	/	/	/	/	/	/	/
Dorsal Raphe	/	/	/	/	/	/	/	/	/	/	/
Median Raphe	/	/	/	/	/	/	/	/	/	/	/
Ventral Tegmental Area	/	/	/	/	/	/	/	/	/	/	/
Nucleus Basalis of Meynert	right	14	−4	−12	3.35	3.32	3	0.000	1.000	0.571	3
Control Pontine ROI	/	/	/	/	/	/	/	/	/	/	/

**MCI** (n. 156)											
Locus Coeruleus	left	−4	−40	−28	3.57	3.49	7	0.000	1.000	0.639	21
Dorsal Raphe	/	/	/	/	/	/	/	/	/	/	/
Median Raphe	/	/	/	/	/	/	/	/	/	/	/
Ventral Tegmental Area	/	/	/	/	/	/	/	/	/	/	/
Nucleus Basalis of Meynert	/	/	/	/	/	/	/	/	/	/	/
Control Pontine ROI	/	/	/	/	/	/	/	/	/	/	/

**AD** (n. 135)											
Locus Coeruleus	right	6	−36	−20	4.41	4.25	8	0.000	0.992	0.463	9
Dorsal Raphe	/	/	/	/	/	/	/	/	/	/	/
Median Raphe	/	/	/	/	/	/	/	/	/	/	/
Ventral Tegmental Area	left	−4	−22	−16	3.78	3.68	13	0.000	1.000	0.618	14
Nucleus Basalis of Meynert	right	14	−4	−12	3.83	3.72	19	0.000	1.000	0.583	19
Control Pontine ROI	/	/	/	/	/	/	/	/	/	/	/

^a^ Peak T value: T value of the most significant cluster of contiguous voxels; ^b^ Peak Z-score: Z-score of the most significant cluster of contiguous voxels; ^c^ Peak cluster Ke: number of voxels of the most significant cluster of contiguous voxels; ^d^
*p* Value Uncorrected; ^e^ FWE = family wise error correction value; ^f^ FDR = false discovery rate correction value (q); ^g^ Total number of voxels outcoming in the ROI including all clusters of contiguous voxels.

### 3.5. Bayes Factors: Parameters of Evidence Strength for the Six ROIs Involved in Attention and Brain Maintenance across the Three Groups

BF_10_ confirmed the disproportional predictive involvement of the LC–NA system observed in the VBM analyses. Bayesian modelling demonstrated that LC volume exhibited the strongest relationship with BrainPAD and TMT-A (with the one exception of the LC—TMT-A correlation in the HC group).

**Table 3 cells-10-01829-t003:** VBM multivariate linear regression analyses for the six ROIs across the three groups (HC n = 395; MCI n = 156; AD n = 135). The results were adjusted for total intracranial volume (TIV), while age and education were entered as continuous variables. BrainPAD values were treated as the continuous dependent variable and the structural scans as the independent variable. For a statistical threshold of *p* < 0.001 FWE corrected (HC) and *p* < 0.001 (MCI and AD), the table reports the significant voxels negatively related to BrainPAD.

Brain Regions	Side	MNI Coordinates			Peak T Value ^a^	Peak Zscore ^b^	Peak Cluster Ke ^c^	*p*-Value Uncorr ^d^	FEW ^e^	FDR ^f^	Total Number of Voxels *p* < 0.001 FWE Threshold ^g^
		*x*	*y*	*Z*							
**HC** (n. 395)											
Locus Coeruleus	left	−2	−38	−22	9.10	inf	106	0.000	0.000	0.000	153
Dorsal Raphe	right	2	−32	−12	8.14	7.81	71	0.000	0.000	0.000	71
Median Raphe	/	/	/	/	/	/	/	/	/	/	/
Ventral Tegmental Area	/	/	/	/	/	/	/	/	/	/	/
Nucleus Basalis of Meynert	/	/	/	/	/	/	/	/	/	/	/
Control Pontine ROI	/	/	/	/	/	/	/	/	/	/	/
**MCI** (n. 156)											total number of voxels for *p* < 0.001 threshold ^g^
Locus Coeruleus	left	−2	−34	−14	4.16	4.04	16	0.000	1.000	0.125	59
Dorsal Raphe	right	2	−30	−8	5.78	5.48	109	0.000	0.021	0.563	109
Median Raphe	/	/	/	/	/	/	/	/	/	/	/
Ventral Tegmental Area	left	−2	−16	−14	3.22	3.16	1	0.001	1.000	0.369	2
Nucleus Basalis of Meynert	right	14	−8	−10	4.15	4.03	12	0.000	0.999	0.093	12
Control Pontine ROI	/	/	/	/	/	/	/	/	/	/	/

**AD** (n. 135)											
Locus Coeruleus	left	−4	−36	−16	5.56	5.25	94	0.000	0.065	0.005	192
Dorsal Raphe	left	−2	−32	−12	4.74	4.54	90	0.000	0.786	0.037	90
Median Raphe	/	0	−34	−22	4.19	4.05	20	0.000	1.000	0.135	20
Ventral Tegmental Area	left	−4	−22	−16	4.49	4.32	9	0.000	0.976	0.067	9
Nucleus Basalis of Meynert	right	14	−6	−12	3.41	3.33	5	0.000	1.000	0.655	5
Control Pontine ROI	/	/	/	/	/	/	/	/	/	/	/

^a^ Peak T value: T value of the most significant cluster of contiguous voxels; ^b^ Peak Z-score: Z-score of the most significant cluster of contiguous voxels; ^c^ Peak cluster Ke: number of voxels of the most significant cluster of contiguous voxels; ^d^
*p* Value Uncorrected; ^e^ FWE = family wise error correction value; ^f^ FDR = false discovery rate correction value (q); ^g^ Total number of voxels outcoming in the ROI including all clusters of contiguous voxels.

Overall, across the three groups, as indicated by BF_10_, the LC likelihood to predict brain maintenance was 19321.07 times more than the null hypothesis, whereas it was 779.29 for the other five ROIs when summed together. Similarly, the LC likelihood to predict attention was 4158.30 times more than the null hypothesis, while it was 241.35 for the sum of the other five ROIs. However, there were notable differences between the groups. Concerning BrainPAD, the MCI group compared to HC and AD showed the strongest evidence for LC (BF_10_ 19303.214), followed by DR (BF_10_ 399.634) and NBM (BF_10_ 339.646). Furthermore, it is noteworthy that there is evidence of absent relationships (support for the null hypothesis) for all other ROIs in the MCI group (BF_10_ < 1 indicating more evidence for the null hypothesis (i.e., no relationship). Similarly, for TMT-A, the LC showed the strongest evidence (BF_10_ 517.357), followed by the DR (BF_10_ 4.219), and for all other ROIs, there were no relationships. In the AD group, the evidence for the LC was also the most substantial (BF_10_ 46.538), but there was also evidence for more distributed involvement of other nuclei as the DR (BF_10_ 7.031), the MR (BF_10_ 2.541) and the VTA (BF_10_ 4.657) supporting BrainPAD. In the same vein, for TMT-A, the strongest evidence was in support of the LC (BF_10_ 3640.710), but strong evidence was also found for the VTA (BF_10_ 124.076) and the NBM (BF_10_ 102.490) but no evidence for the other ROIs. In the HC group, the magnitude of the results was less pronounced but followed the same pattern. The LC showed the strongest evidence supporting BrainPAD (BF_10_ 142.324) followed by the DR (BF_10_ 22.466), and no evidence was found for all the other ROIs. In contrast to the clinical groups, no evidence for the ROIs and TMT-A was found in the HC, with the exception of the DR (BF_10_ 9.117). BF_10_ values are reported specifically for the three groups in Table 4 along with Pearson’s correlation coefficients. BF_10_ values were covaried for age, education and total intracranial volume.

### 3.6. Brief Summary of the 3rd, 4th, 5th and 6th VBM Analyses

The other VBM branches of analyses are reported in Appendix A (extended methods and results sections). A brief summary is provided here.

The third and the fourth branches of VBM analyses examined the extent to which education and the degree of occupational cognitive demand were predicted by the six ROIs. In summary, absent or negligible findings were observed for education (3rd VBM branch) and occupational cognitive demand (4th VBM branch). For further details, see Appendix A.

The fifth branch of VBM analyses investigated how the factor group (HC, MCI, AD) affected the volumetric variations within the neuromodulatory subcortical systems. The main areas where the MCI showed decreased volume compared to HC were the DR and VTA. By contrast, there was no LC volume difference in MCI vs. HC group. AD patients showed decreased volume compared to MCI in the LC and in the VTA areas. Consistent with the previous literature [25,27,28,29,32,81,83,131,132,133], the main effect of group on volume reduction was observed in change in the NBM, the DR, the VTA and the LC (listed in order of statistical power). These areas showed the greater deterioration across the different stages of cognitive decline. For further details, see Appendix A along with Figure 6 for spatial resolution.

Since the literature concerning LC gender differences is controversial [25,34,56,117,134], the sixth branch of VBM analyses explored the differential volumetric variation of the neuromodulators due to gender at different stages of cognitive decline. Negligible differences between gender were observed. These differences became weaker or disappeared when covariates (age, TIV and education) were removed from the models. For more details, see Appendix A.

## 4. Discussion

The present study conducted a volumetric analysis of subcortical nuclei in healthy older controls (HC, n = 395), patients with mild cognitive impairment (MCI, n = 156) and Alzheimer’s disease (AD, n = 135). We hypothesized that structural integrity of the locus loeruleus (LC) would be particularly important for brain maintenance and cognitive reserve in the context of age and disease, due to the known neuroprotective effects of the LC–NA system [1,2]. As anticipated, we observed a systematic reduction in attention performance and biological brain maintenance (BrainPAD) across the HC, MCI and AD groups. We also observed a general reduction in the volume of subcortical nuclei at different stages of the cognitive decline. Compared to the other subcortical nuclei, LC volume was most extensively associated with the degree of brain maintenance across the three groups. However, only in MCI patients was the relationship between brain maintenance and attention performance mediated by LC volume, suggesting a unique compensatory role of noradrenergic neuromodulation for MCI patients. Although these findings suggest a significant role for the LC–NA system in brain maintenance, we found negligible or inconsistent associations between the subcortical ROI volumes, BrainPAD scores and indices of cognitive reserve (i.e., education level and occupational status). However, education level did show a positive association with overall brain volume (in the HC and MCI groups), suggesting it might be more closely associated with brain development and expansion rather than later life brain maintenance. Few studies have investigated the relationship between LC volume and attentional performance in older adults [34,49,68]. The present study utilised the Trail Making Test A (TMT-A) as a putative measure of visual attention speed and found that greater LC volume in MCI and AD patients was associated with faster attention performance. A ceiling effect of TMT-A performance in the HC group gave limited variability for detecting an association with LC volume. However, a pattern of reduced and more heterogeneous TMT-A performance in both MCI and AD group revealed relationships with brainstem nuclei that may counteract progressive cortical decline seen in these patients. Histopathological evidence shows that in the early stages of neurodegeneration, the metencephalic areas appear to be less affected than cortical areas [25,26,50,51,52,53,135,136]. It is possible that the degree of structural integrity within these metencephalic regions provides an important neuromodulatory and compensatory role in the face of declining cortical function [1,2,17,21,54,55,61]. Indeed, although the LC shows markers of neurodegeneration years before cognitive decline [26,50], neuronal cell loss in the LC is most appreciable in advanced stages of neurodegeneration rather than in the earlier stages (evidence reported from Braak stages III-IV) [25,26,50,56]. Therefore, throughout the pathological course, the volumetric and functional resilience of the LC (including the maintenance of the surrounding LC-peri dendritic and the epi and sub-coeruleus areas [14,104,105]) may provide a supporting and compensating role in the face of the beta-amyloid pathology occurring in the cortex [16,17,21,26,50,104,136]. Interestingly in this regard, a recent work by Bachman et al. [137] found positive relationships between the volume of the LC and surface measures in cortical regions including the rostral medial frontal cortex, which has a high density of noradrenergic varicosities [92].

In MCI patients, we found that the structural variation of the LC was unique among the six subcortical ROIs in predicting visual attention performance. Specifically, Bayesian analyses demonstrated that there was substantial evidence in support of LC involvement relative to all other ROIs. These analyses strengthened the specificity of the LC–NA system over the other neuromodulators’ seeds and outlines its remarkable role in brain health. Widespread transmission of NA from the LC via diverse cortical efferent projections has been shown to significantly affect attention performance [54,99,138,139,140,141,142]. In the context of MCI, these findings suggest that neuromodulation of the LC–NA system, in particular, may help support the diminishing efficiency of cortical attention networks in this group. In AD patients, several relationships were observed in which greater volumes of the LC, VTA and NBM were associated with faster visual attention performance. Due to the more severe cortical neurodegeneration in AD, it is conceivable that further compensatory support from both ascending catecholaminergic nuclei, including dopaminergic (VTA) and noradrenergic (LC) regions as well as activation of cholinergic basal forebrain nuclei (NBM), would be necessary to support declining cortical function [143]. Indeed, the basic requirements of the TMT-A are exploratory behaviour and selective attention [89,90,98], which are modulated by the catecholaminergic and cholinergic systems, respectively [1,2,14,59,143,144].

A seminal study by Wilson and colleagues [31] demonstrated that greater LC neural density was associated with reduced cognitive decline and increased reserve from an older adult cohort. In the present study, we found LC volume to be a reliable predictor of brain maintenance in each of the three groups. This pattern of results is consistent with Robertson’s noradrenergic theory of reserve [1,2], hypothesizing that protracted activation of the LC–NA system across the lifespan can enhance brain reserve through neuroplastic change, as well as through the neuroprotective effects of reduced neuroinflammation [23,40,41,42,145] and increased BDNF production [43,44,45,46,47,48].

Given the high sensitivity of predicted age discrepancy measures to brain deterioration [10,146,147,148], the universal relationship between BrainPAD and LC volume could indicate that the LC–NA system is a key driver of brain reserve, which is shaped through a more active and efficient attentional system [1,2]. Our findings are consistent with the evidence of post-mortem and neuroimaging studies relating greater LC volume to indices of reserve and to reduced neurodegeneration [18,19,20,21,31,145]. In addition to the LC volume, the observed relationship between DR volume and BrainPAD may also suggest a marked serotoninergic involvement in brain maintenance, particularly in MCI. Serotoninergic deterioration in neurodegenerative diseases has been reported [26,50,131,149,150,151], and degeneration of the DR is well documented [25,52,53,150,152]. However, DR is not only a serotoninergic seed, but a multifunctional nucleus with significant non-serotoninergic pathways [153,154]. Although 70% of the DR neurons contain serotonin, the remaining 30% include other neurotransmitters (such as catecholamine [155,156]) and NA transporters (NET) [157]. Early research [158] has reported that 38% of the amount of the noradrenaline found in the LC is found in the DR (LC: 1.22 mg/g tissue mean; DR: 0.47 mg/g tissue mean). Therefore, the relationship between BrainPAD and the DR volume must also be considered in the context of the degeneration of the LC–NA system, particularly given the absence of evidence for the MR and BrainPAD. The LC has been shown to exhibit early signs of neurodegeneration before the DR nucleus [25], and patients with dementia exhibit significantly reduced noradrenaline transporters (NET) in the LC but not in the DR [55]. Together these findings suggest that noradrenergic neurons within the DR might provide compensatory support for the broader LC–NA system [54,55,103,104]. Accordingly, the resilience of the DR as a compensatory nucleus to the LC–NA system decay may, in part, explain the observed relationships between DR volume and BrainPAD in the present study.

In the present study, we also compared the influence of each of the five neuromodulator seed ROIs in mediating the relationship between brain maintenance and attentional efficiency. In keeping with Robertson’s noradrenergic theory of cognitive reserve, we hypothesized that greater LC volume would be a key mediator in this relationship. However, only in MCI patients did LC volume account for the association between BrainPAD and attentional performance, suggesting that at prodromal stages of neurodegeneration greater LC integrity in the context of declining cortical systems may provide critical support for attention processes. By contrast, healthy older adults may have sufficient brain maintenance without additional neuromodulatory support from the LC and, in AD patients, pronounced LC degeneration compared to MCI and HC [31,32,55,56,69] may undermine its compensatory role in later stages of neurodegeneration. Although the DR nucleus has significant noradrenergic expression, we did not find a direct relationship between DR volume and attention performance, nor did DR volume mediate the association between brain maintenance and attention performance. In the current study, putative indices of cognitive reserve, education and occupational status were strongly related to each other but not associated with BrainPAD scores, subcortical ROI volumes or attention performance. Although previous findings [159] have reported an association between higher education and younger brain age, more recent research has not yielded such a relationship [160]. Alternatively, it is plausible that factors such as education and occupation are more directly related to actual reserve built through developmental plasticity in the early stages of life, and indexed by total intracranial volume (TIV), and are therefore more indirectly related to brain maintenance in the later stages of life [8].

## 5. Limitations

TMT-A is a valuable clinical tool to assess higher cognitive functions and it is widely used for its sensitivity to basic attentive efficiency, but it is a limited measure for assessing overall noradrenergic contributions to cognition. The cortical influence of NA is global and complex, and a single measure lacks the necessary richness to capture this entirely. Indeed, it is acknowledged how the LC–NA system plays a crucial role in memory [33,68,69,70,131,145] and in other higher cognitive functions distinct from visuo-spatial attention [1,2,17,19,59]. Nevertheless, our findings do suggest a markedly predominant role for the LC in attention and brain maintenance, particularly given the absence of a relevant effect for other neuromodulatory control nuclei. The sample size of the HC group (n = 395) is considerably larger than both MCI (n = 156) and AD (n = 135); therefore, uncorrected results for BrainPAD in MCI and AD might be driven by the reduced size of these groups.

The retrospective broad nature of the measures of sociodemographic indices of reserve, particularly job complexity, may not offer information precise enough to capture the effects of this variable. The job classification in ranks according to the supposed cognitive demand can only reflect hypothetical contingencies related to the occupation.

In addition, despite that neuropsychological and volumetric measures showed clear and systematic differences across the groups (as expected based on the standard diagnostic criteria used by the ADNI), it should be considered that subsequent modifications to the diagnostic criteria may yield some miss-classification when compared to the initial baseline rollover of the ADNI3 phase [161,162,163], since MCI criteria for diagnosis are constantly being reviewed and scrutinized [164]. However, TMT-A performances of HC group are within the normative values for their age and education, suggesting that there is no evidence of behavioural impairment, at least based on normative scores for the TMT-A, suggesting no evidence of impairment in this group [165].

Lastly, the methodological limitations of VBM analyses in cross-sectional studies should be taken into account while considering these findings. However, within these limitations, the utmost rigour was employed in defining the subcortical regions, while acknowledging the comprehensive literature on the subject. The processing pipeline was also strict in terms of quality assurance: an attempt was made to account for all possible confounding covariates. Additionally, opposite hypotheses have been tested as well. Despite these limitations, the findings of the current study are consistent both with histopathological and neuroimaging studies on noradrenergic, serotoninergic, cholinergic and dopaminergic systems in the context of neurodegenerative diseases [25,27,28,29,32,81,83,131,132,133].

## 6. Conclusions and Clinical Implications

The findings of this study are consistent with the vast literature on the decay of the LC–NA system in neurodegenerative diseases, and with a growing number of studies showing how the LC–NA system is a crucial mediator of reserve both in healthy and pathological ageing, as postulated by Robertson. This work is among the largest MRI studies carried out on the noradrenergic system, counting more than 250 VBM analyses on 686 subjects. Additionally, it is the first work investigating the comparative relationships between an objective measure of biological brain health and the integrity of neuromodulatory subcortical systems. This work extends the knowledge of the role of the LC–NA system in the neurobiology of cognitive decline and also as potential in-vivo biomarker of neurodegenerative diseases.

The relationships identified in this study highlight the need to target therapeutic approaches, which focus on enhancing the function and the structural integrity of the LC–NA system. As proposed by Robertson, early prevention strategies which focus on upregulation of the noradrenergic system in ageing and dementia may yield important clinical benefits. This is possible through cognitive stimulation via attentional training (involving sustained attention and working memory capacity) [1,2,21]. Indeed, visual attention and working memory are domains underpinned by the noradrenergic system [1,2,21,59,91]. Cognitive interventions involving these domains performing exercises of mental flexibility, problem solving and visual search might stimulate the LC–NA system enhancing cognition [1,2]. Short cognitive interventions based on this approach showed cognitive improvements related to frontal areas with predominant presence of noradrenergic receptors [92,166].

In addition, noradrenergic drugs which increase LC activity and cognitive performances might be beneficial [57,167,168,169]. Studies using noradrenergic drugs showed that interventions ameliorated AD-like pathology and partially re-stored noradrenergic tone in humans [104,167,168]. However, the benefit of noradrenergic drugs is controversial, and side effects should be considered while evaluating this approach (see Holland et al. (2021) for a review) [59].

Another potential intervention might involve physical exercise, which is among the main preventing factors of dementia [13]. Physical exercise is known to help maintain proper upregulation of the LC–NA system. It has been shown that physical exercise boosts NA release [39] and activates LC–NA system, which is linked to improved cognition in healthy and MCI populations [21,58,170].

Diet may also play a supportive role within the noradrenergic system as a preventing factor of neurodegenerative diseases. Indeed, poor and wrong diet-styles and their related neuroinflammatory consequences [171,172,173] are associated with worse prognosis and higher neurodegeneration risks [13,171,172,174,175]. Interventions on diet may be beneficial to restore potential harmful micro and macro nutrient insufficiencies according to current guidelines for prevention [13,176]. In particular, as observed in regards of the cholinergic system decay with dementia, a higher level of choline intake and supplementation has been shown to improve cognitive performances in both healthy people and patients [177,178,179]. Likewise, tyrosine intake and supplementation in healthy elderly and in MCI patients offer another possible conceivable way of ameliorating the LC–NA system degeneration and cognitive decline [51,55,180,181,182,183,184]. It is worth mentioning that several studies in the past linked tyrosine supplementation to increased cognitive performances in healthy subjects [185,186,187,188,189]. Furthermore, a recent study by Kühn et al. (2019) [180] found in 1724 healthy individuals that the average dietary tyrosine intake in grams related to better cognitive performances in working memory, episodic memory and fluid intelligence.

Finally, other potential interventions may involve breathing practices such as meditation, pranayama and breath-control, which are thought to involve the LC–NA system [190,191]. Some studies on long-term meditators have indeed reported increased brain volume in critical areas affected in dementia, including the brainstem, along with better cognitive efficiency [165,192,193,194,195], after as a little as 5–10 h of practice [196,197].

As an ultimate conclusion, it should be considered that the current study did not directly investigate the upregulation of the LC–NA system but explored cross-sectionally the volumetric integrity of the LC–NA system in comparison with the other main neuromodulator seeds. These investigations highlight the relevance of the LC–NA system in attentional domain and in the biological component of reserve, linking brain health to greater integrity of the LC–NA system. Although greater LC volume may ensure greater availability of NA reported in multiple studies [1,2,17,18,20,21,22,25,26,28,30,33,36,47,49,50,51,52,53,54,55,56,57,58,59,60,61], it is important to acknowledge that further integration of the relationship between structure and function of the LC is warranted.

## Figures and Tables

**Figure 1 cells-10-01829-f001:**
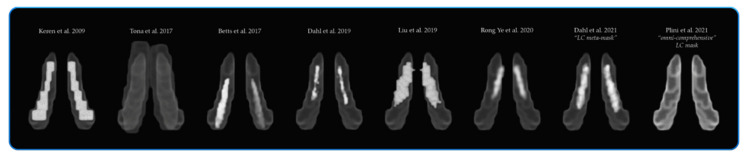
The spatial resolution of the new created symmetrical “omni-comprehensive” LC mask in comparison with the previously published LC MRI atlases and masks.

**Figure 2 cells-10-01829-f002:**
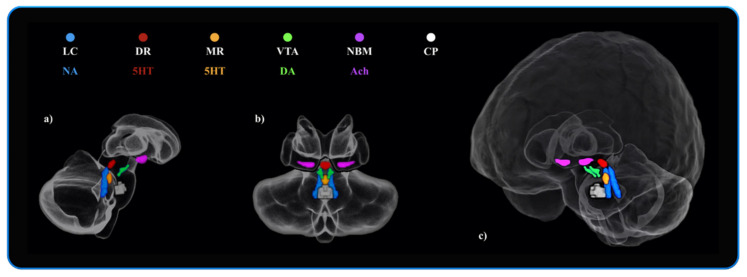
The neuromodulators’ seeds and the neuromodulator-free control region investigated in the VBM analyses. Blue: Locus Coeruleus (LC—“omni-comprehensive”)—Noradrenaline (NA)—(714 voxels). Red: Dorsal Raphe Nuclei (DR)—Serotonin (5HT)—(174 voxels). Orange: Median Raphe Nuclei (MR)—Serotonin (5HT)—(108 voxels). Green: Ventral Tegmental Area (VTA)—Dopamine (DA)—(252 voxels). Purple: Nucleus Basalis of Meynert (NBM)—Acethylcholine (Ach)—(492 voxels). White: Control Pontine ROI (“neuromodulator-free” region)—(906 voxels). All the ROI binary masks were symmetrical with a 1 mm3 isotropic voxel size, and there were no overlapping boundaries between the masks. Image (**a**) displays the anatomical localization of the six ROIs in sagittal view of the Cerebellum, Brainstem and Diencephalon. Image (**b**) displays the coronal view. Image (**c**) shows a fronto-lateral view of the whole brain and the anatomical localization of the six ROIs.

**Figure 3 cells-10-01829-f003:**
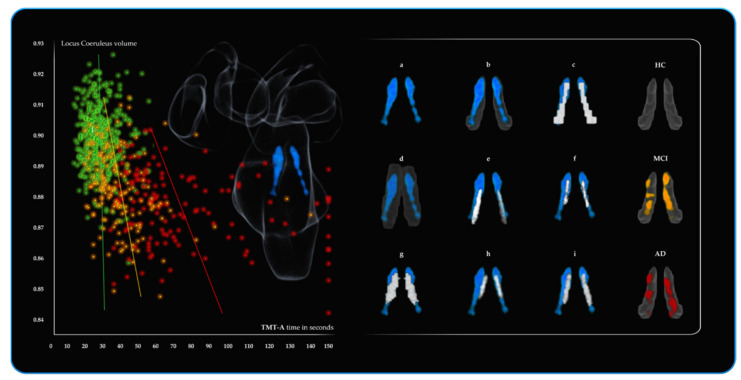
Results from the VBM multivariate linear regression analyses performed in CAT12 for the three groups (HC, MCI, AD). The results are covaried for total intracranial volume (TIV), age and education. The scatterplot displays the relationship between Locus Coeruleus (LC) volume (y axis) and TMT-A time in seconds (x axis) for the three groups: green (HC, n.395), orange (MCI, n.156), red (AD, n.135). On the x axis (TMT-A) are the seconds required to complete TMT-A. More seconds spent in completing TMT-A mirror a slower visuo-spatial cognitive processing related to the LC decline. The systematic decline of the LC volume across the three groups is related to a slower visuo-motor attentional performance. On the left portion of the figure, blue shows the average LC results for the three groups (n = 686, *p* < 0.001) on a 3D fronto-lateral view of the Brainstem and the Diencephalon. On the right portion of the figure are the 3D reconstructions (displayed in the MNI152 space) of the results in comparison with previously published LC atlases and masks. (**a**) average LC result; (**b**) average LC result is shown in the LC “omni-comprehensive” mask; (**c**) Keren et al. (2009) [102,107]; (**d**) Tona et al. (2017) [108]; (**e**) Betts et al. (2017) [109]; (**f**) Dahl et al. (2019) [33]; (**g**) Liu et al. (2019) [110]; (**h**) Rong Ye et al. (2020) [111]; (**i**) LC meta-mask by Dahl et al. (2021) [70]. The last column on the right shows the regions of the LC mask negatively related to TMT-A performances for the three groups considered separately (*p* < 0.01): HC *n* = 395 (no results), MCI *n* = 156 and AD *n* = 135.

**Figure 4 cells-10-01829-f004:**
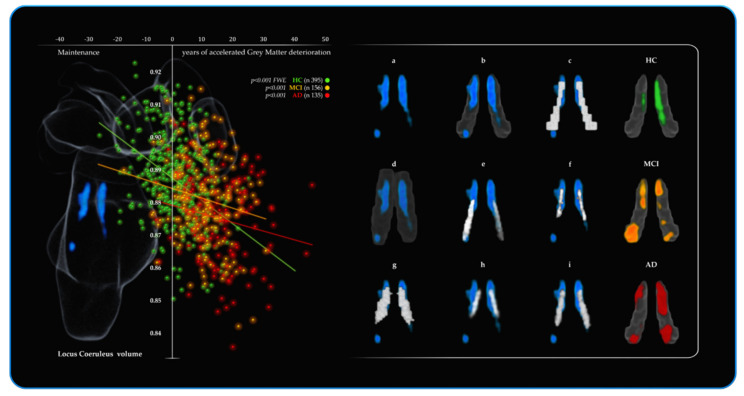
Results from the VBM multivariate linear regression analyses performed in CAT12 for the three groups (HC, MCI, AD). The results are covaried for total intracranial volume (TIV), age and education. The scatterplot displays the relationship between Locus Coeruleus (LC) volume (y axis) and BrainPAD (x axis) for the three groups: green (HC, n.395), orange (MCI, n.156), red (AD, n.135). On the x axis (BrainPAD) are the years of discrepancy between chronological age and the actual biological brain age based on the degree of Grey Matter deterioration. Negative values of BrainPAD mirror a better GM brain maintenance corresponding to a “younger brain age” than the chronological age. Positive values of BrainPAD mirror the accelerated brain’s GM deterioration, namely, positive values correspond to a worse biological maintenance, and to an older brain than the chronological age. The systematic decline of the LC volume across the three groups is related to a progressive accelerated GM deterioration. Indeed, HC values are shifted more towards the upper-left portion of the graph (negative values of BrainPAD and greater LC volume), whereas MCI and AD are more shifted towards the lower-right portion of the graph (positive values of BrainPAD and lower LC volume). On the left portion of the figure, blue shows the average LC results for the three groups (n = 686, *p* < 0.00001 FWE) on a 3D fronto-lateral view of the Brainstem and the Diencephalon. On the right portion of the figure are the 3D reconstructions (displayed in the MNI152 space) of the results in comparison with previously published LC atlases and masks. (**a**) average LC result; (**b**) average LC result is shown in the LC “omni-comprehensive” mask; (**c**) Keren et al. (2009) [102,107]; (**d**) Tona et al. (2017) [108]; (**e**) Betts et al. (2017) [109]; (**f**) Dahl et al. (2019) [33]; (**g**) Liu et al. (2019) [110]; (**h**) Rong Ye et al. (2020) [111]; (**i**) LC meta-mask by Dahl et al. (2021) [70]. The last column on the right shows the regions of the LC mask negatively related to BrainPAD values for the three groups considered separately: HC n = 395 (*p* < 0.001 FWE), MCI n = 156 (*p* < 0.001) and AD n = 135 (*p* < 0.001).

**Figure 6 cells-10-01829-f006:**
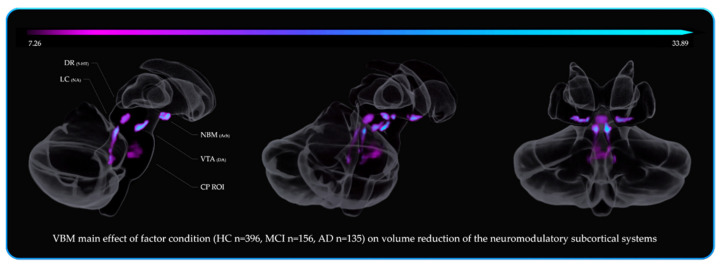
The main effect of factor condition (HC n = 395, MCI n = 156, AD n = 135) on the six ROIs. The VBM results were thresholded for *p* < 0.05 FWE (corrected) and adjusted for total intracranial volume, age and education. The significant clusters of voxels are shown on a 3D reconstruction of the Cerebellum, the Brainstem and the Diencephalon (sagittal to coronal view from left to right). The color bar represents F value ranges and reflects the significance of volume reduction across the three groups. The main volume reduction occurred in the Nucleus Basalis of Meynert (NBM) 35 voxels F 33.89, in the Ventral Tegmental Area (VTA) 65 voxels F 23.79, in the Dorsal Raphe (DR) 132 voxels F 28.66 and in the Locus Coeruleus (LC) 64 voxels F 21,46. See Appendix A with Figure 2 for further details.

**Table 1 cells-10-01829-t001:** Sociodemographic, neural indices and neuropsychological characteristics of the HC, MCI and AD groups. Key: TIV, total intracranial volume; BrainPAD, Brain predicted age discrepancy (biological maintenance index); TMT-A. trail making test part A time in seconds. (* *p* < 0.05; *******
*p* < 0.001 Bonferroni correction).

	Sociodemographic								Neural Indices				Neuropsychological	
	Gender		Age ***		Education ***		Occupation *		TIV		BrainPAD ***		TMT-A ***	
Groups	M	F	Mean	SD	Mean	SD	Mean	SD	Mean	SD	Mean	SD	Mean	SD
*Ranges*			*min*	*max*	*min*	*max*	*min*	*max*	*min*	*max*	*min*	*max*	*min*	*max*
HC (n = 395)	168	227	73.45	7.21	16.96	2.21	6.21	1.72	1408.16	146.28	1.59	7.69	30.64	8.85
			*56*	*95*	*11*	*20*	*2*	*8*	*1082*	*1844*	*−22.01*	*37.46*	*13*	*63*
MCI (n = 156)	90	66	75.51	8.10	15.98	2.76	5.78	1.85	1426.12	142.78	5.45	9.00	39.07	17.86
			*55*	*97*	*8*	*20*	*2*	*8*	*1060*	*1884*	*−21.23*	*26.21*	*18*	*150*
AD (n = 135)	74	61	76.61	8.43	15.64	2.54	5.93	1.84	1385.79	167.47	13.32	9.13	61.61	36.59
			*55*	*95*	*11*	*20*	*2*	*8*	*1046*	*1785*	*−6.46*	*42.33*	*19*	*150*

**Table 4 cells-10-01829-t004:** The Bayes factors (BF_01_) for the three groups showing the strength of relationships between the six ROIs, BrainPAD and TMT-A. ROIs average values were corrected for age, education and total intracranial volume.

		Locus Coeruleus	Dorsal Raphe	Median Raphe	Ventral Tegmental Area	Nucleus Basalis of Meynert	Control Pontine ROI
**HC (n = 395)**							
BrainPAD	Pearson’s r	−0.197 ***	−0.292 ***	−0.107	−0.074	−0.107	−0.109
	**BF_10_**	142.324	22.466	0.608	0.185	0.599	0.643
TMT-A	Pearson’s r	−0.082	−0.158	−0.025	−0.045	0.085	−0.042
	**BF_10_**	0.236	9.117	0.071	0.094	0.257	0.089
**MCI (n = 156)**							
BrainPAD	Pearson’s r	−0.385 ***	−0.321 ***	−0.039	0.147	−0.318 ***	0.007
	**BF_10_**	19303.214	399.634	0.113	0.523	339.646	0.101
TMT-A	Pearson’s r	−0.326 ***	−0.219	0.001	0.019	0.029	0.007
	**BF_10_**	517.357	4.219	0.100	0.103	0.107	0.101
**AD (n = 135)**							
BrainPAD	Pearson’s r	−0.297 **	−0.249	−0.217	−0.305	−0.132	0.100
	**BF_10_**	46.538	7.031	2.541	4.657	0.340	0.209
TMT-A	Pearson’s r	−0.384 ***	−0.119	−0.041	−0.319 ***	−0.315 ***	−0.5224
	**BF_10_**	3640.710	0.272	0.120	124.076	102.490	0.108

** BF_10_ > 30, *** BF_10_ > 100.

## Data Availability

***** Data used in preparation of this article were obtained from the Alzheimer’s Disease Neuroimaging Initiative (ADNI) database (adni.loni.usc.edu accessed on 15 January 2021). As such, the investigators within the ADNI contributed to the design and implementation of ADNI and/or provided data but did not participate in analysis or writing of this report. A complete listing of ADNI investigators can be found at: http://adni.loni.usc.edu/wp-content/uploads/how_to_apply/ADNI_Acknowledgement_List.pdf (accessed on 15 January 2021).

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
