# Peer review of "Examining the Role of the Noradrenergic Locus Coeruleus for Predicting Attention and Brain Maintenance in Healthy Old Age and Disease: An MRI Structural Study for the Alzheimer’s Disease Neuroimaging Initiative"

_cells, 2021, doi:10.3390/cells10071829_

Round 1

Reviewer 1 Report

The study of Plini et al. focused on the role of locus ceruleous and noradrenergic system in general as key mediator underpinning the neuropsychology of cognitive reserve. The topic is relavant since great attention is always given to cortical areas involved in neurodegenerative disorders leading to cognitive decline, whilst subcortical areas of CNS are often under-investigated. The study is well-designed and the manuscript is well-written.

As single suggestion, this Reviewer thinks that the Authors should stress more in the last paragraph clinical relavance and consequence of their findings, in terms of both pharmacologycal and non-pharmachologycal interventions. 

Author Response

Dear Reviewer 1, please find attached the comment and revisione made following your suggestions.

Reviewer 2 Report

This paper uses the publicly available ADNI data to examine voxel based morphometry (VBM) estimates of volume from ROIs targeting 6 subcortical nuclei and how they relate to attentional performance on one test (TMT-A) and brain grey matter structural integrity (using a ‘BrainPad’ score from a brain-age algorithm). The authors found that lower volume of voxels in the LC were associated with poorer TMT-A performance within the MCI group (consistent with prior findings using the same dataset) as well within the AD group. There were also some associations with other nuclei. However, none of these results held up to multiple comparison correction. There were more robust relationships between ROI voxel volume estimates and BrainPad scores, at least for healthy controls (HC) for LC and DR. DR but not any other nuclei also survived FWE correction for the MCI group, and none survived FWE correction for AD. LC VBM volume estimates mediated the relation between BrainPAD and TMT-A in the MCI group but not in the HC or AD groups. But here as elsewhere, a number of tests were run and it is not clear whether there was multiple-comparison correction for this mediation effect.

This is an ambitious effort to characterize a complex set of systems. The paper has various strengths such as the thorough literature review, inclusion of this broad set of neuromodulatory systems, and use of a control pons ROI. However, in order to address their hypotheses, the authors conducted a lot of statistical tests. Most effects did not survive multiple comparisons correction. Those that did were the relationship between volume estimates in some subcortical nuclei and the brain age estimate derived from cortical grey matter within the healthy controls and within the MCI group. That there is a relationship between structural integrity in subcortical and cortical brain regions is not surprising.

Instead, what the authors seem to be arguing is that the findings indicate some unique role of the LC relative to the other nuclei. However, whether the relationship between the LC and attention or cortical grey matter is stronger than the equivalent relationship with other nuclei is never tested. Throughout the paper, the authors seem to assume that if the relationship between ROI A and an outcome measure is significant whereas the relationship between ROI B and the same outcome measure is not significant, this means that ROI A is more influential than ROI B in determining that outcome measure. However, this assumption is not correct. For instance, an r value of .4 could be significant whereas another r value of .39 could be just below statistical significance and the differences in the significance of these two p values would not indicate that one correlation has a significantly stronger relationship than the other. Specific statistical tests are necessary to draw these sorts of comparative conclusions. 

Here are two examples of where the authors make this type of “significance vs non-significance = a significant difference in relationships in the two scenarios” assumption: 

1. p. 15: “only in MCI patients was the relationship between brain maintenance and attention performance mediated by LC volume suggesting a unique compensatory role of noradrenergic neuromodulation for MCI patients.” – to draw this conclusion about the uniqueness of the LC role in MCI, statistical comparisons between the strength of mediation in the different groups would be necessary. 

2. p. 17: “In the present study, we also compared the influence of each of the five neuromodulator seed ROIs in mediating the relationship between brain maintenance and attentional efficiency.” – this statement is not accurate; including data from the five seed ROIs in a multiple regression does not provide a direct comparison of their influence. No statistical test was run that explicitly contrasts the strength of the relationships of these seed ROIs with brain structure or attentional performance.

Similarly, in the discussion the authors claim their findings “suggest an exclusive role for the LC in cognitive maintenance” (page 17 line 606), yet they found relationships between their BrainPAD metric and Dorsal Raphe volume, which contradicts the exclusivity of the relationship between cognitive maintenance and LC volume.

Other major points

The authors focus on the influence of these subcortical nuclei on just one of multiple cognitive measures available in this dataset, the TMT-A. This selective focus requires some additional justification given that measures of LC MRI structural contrast have also been found to relate to memory performance (Dahl et al., 2019; Hammerer et al., 2018) and an explicit comparison revealed that a unidimensional model in which the LC contrast relates to a single factor representing all cognitive and behavioral measures available was a better fit to the Cam-CAN results than a two-factor model in which the LC contrast is better associated with putatively LC-related functions (Liu et al., 2020). If the authors stick to focusing just on attention, they may want to consider including a measure of verbal attention (e.g., auditory verbal learning test trial 1) also available in this dataset.

Related to this, the authors make the claim that the LC findings are specific to attention, but there are no comparative analyses with other cognitive domains (memory, language, executive function, global cognition) that would be necessary to show that LC volume is uniquely associated with attention.

The authors alternate between discussing LC volume and LC MRI contrast throughout paper without explicitly discussing how these measures differ. There is an important distinction to make regarding LC contrast as captured with a specialized scan (e.g., T1-fast spin echo), which is not performed in the present study. 

The authors present their third objective (relationships between LC volume and education/occupation) as a main objective in the Introduction (page 5 line 158), but analyses are relegated to supplementary materials (page 5 line 172). Why was this done if it was a central component of the paper? The authors should consider including these findings in the main results.

The theoretical framework underlying the hypotheses is that of cognitive reserve. However, to date there is no compelling evidence that cognitive reserve slows down the rate of structural brain decline. In fact, a recent paper provides evidence that education has no effect on structural brain decline (Nyberg, Magnussen, Lundquist et al., 2021).

Due to the known heterogeneity of Alzheimer’s disease, it would be helpful to characterize the biomarker status of participants. Did the authors consider dividing up participants by biomarker (i.e., ABeta and tau) positivity, using either CSF or PET? Perhaps the observed relationships are driven by individuals with higher (or lower) AD-related pathological burden?

The authors used the ADNI diagnoses of individuals as CN, MCI, or AD. The ADNI sample is known to have misclassified individuals who are cognitively normal as MCI and this has been widely reported (Edmonds et al., 2014, Alzheimer’s & Dementia; Edmonds et al., 2016, Journal of Alzheimer’s Disease). Alternative approaches include a new diagnostic framework (Bondi et al., 2014, Journal of Alzheimer’s Disease) or cluster analysis (Clark et al., 2013, JINS; Edmonds et al., 2014, Alzheimer’s & Dementia) to resolve this issue. Did the authors consider this approach to resolve diagnostic issues? Group differences in all analyses would be affected by any diagnostic misclassification.

Suggesting that misclassifications were indeed an issue in this analysis is that in the supplemental materials it is reported that the NbM showed no significant effects of volume across the HC, MCI and AD groups, in contrast to a previous investigation that used CSF markers of AD to delineate groups (Fernandez-Cabello, Kronbichler, Van Dijk et al., 2020).

The authors should be cautious in their interpretation of results. A major claim is made in the Abstract that the present study results suggest that “upregulating the noradrenergic system” will yield clinical benefits. While this may be the case in general, the present study was entirely volumetric in nature and does not speak to up- or down-regulating the noradrenergic system. There was no intervention performed to increase LC activity or facilitate NA release, and rather the study examined the underlying tissue volume in the chief noradrenergic nucleus and associations with education and occupation. The authors should consider editing the conclusions that can be drawn from this and posit that future studies may explore upregulation of the NA system, rather than the present one.

The LC ROI is based on four of the six previously published LC masks. The masks from Liu et al. (2019) and Dahl et al. (2019) were not included; however, these were the two masks based on the largest samples and a recent preprint found that these two masks had the best specificity/sensitivity combinations out of the six published to date (Dahl, Mather, Werkle-Bergner, et al., see Figure 2). Why were only the lower performing masks included in the current investigation?

Regarding the CAT12 quality control procedure for MRI images, the authors report what percentage of scans were below 80% (page 5 line 192), but critically do not report what percentage or how many scans were discarded during the manual qc step (page 5 line 181) or discarded due to being below 70% (page 5 line 189). This was not found either in the main text or in the supplementary materials and should be reported.

In supplementary material the authors mention that in VBM analyses, both the WM segmentation alone and the GM+WM segmentations were analyzed. However, it is not reported which of these two was ultimately used for all the results reported in the main text. The authors should specify which segmented maps were used for analyses and reported results in the main text, as well as whether findings differed when performing analyses with WM alone versus with GM+WM segmentations.

In the VBM regression models, the authors do not specify what the specific search volume was in each analysis. For each VBM model, was the LC (or other ROI or interest) included as an explicit mask? i.e., was the search volume for each analysis only within the corresponding ROI? If so, the authors should specifically mention this or their exact approach, as it is crucial to understanding the analysis that was performed.

Related to the above point, on page 7 Line 279 the authors state that “A further step in the analyses was to indicate the LC mask as inclusive in order to isolate the LC involvement in the model”- what are the authors referring to in this analysis? How is this different from the analysis described immediately before in the section 2.1.4? This may be clarified in the authors clearly describe their approach to running VBM regression models. 

One way to further determine the specificity of the LC in all analyses would be to search within the entire brainstem, examine which voxels correlated with their various measures of interest (attention, education, occupation), then check to see what ROIs overlap with the resulting significant clusters (if any) in the brainstem or pons. At present, it is unclear if the authors performed such an analysis or not, and such an analysis may strengthen their LC specificity claims.

The authors include TIV, education, and age as covariates but not gender- why was this done? There are known gender/sex differences in the LC/NA system (Ycaza Herrera et al., 2019, Progress in Neurobiology)  

Given the apparently skewed distribution of the trail making test, did the authors consider log transformation or another transformation of the data prior to analyses? 

The significance thresholds reported vary throughout the results and are not clearly stated. The thresholds used in VBM analyses should be stated upfront in the methods, and then clearly stated each time a finding is reported whether it was significant with or without FWE correction.

The authors plotted LC volume measures in Figures 2 and 3, but these values are not reported anywhere in the main text or in Table 1, nor in the supplementary materials (though they do report the group differences in LC volume). The authors should consider adding in LC volume (and all other nuclei volume) into Table 1 or elsewhere. 

Figures 2 & 3 would benefit from reorganization and modification for clarity. In Figure 3, for example, the main VBM finding is displayed underneath a scatterplot, which makes it very difficult to view the findings. Additionally, there are box and whisker plots in the scatter plot that do not appear to correspond to any meaningful values, and their appearance is not described in the figure caption. If the box and whiskers are meant to correspond to the means for each group, that does not appear to be the case based on the distribution of the scatter plots.

The authors mention that the TMT-A test is related to “exploratory behavior” in the Discussion (page 16 line 545)- this is a vague and undefined cognitive process and the authors do not cite any papers tying TMT-A to “exploratory behavior”. This should be further explained.

Overall, the manuscript would benefit from copyediting for grammar, punctuation, and unnecessary verbiage. There are numerous instances of redundant text throughout the manuscript and editing to ensure conciseness would improve the manuscript’s readability.

Minor points

p. 3: should also cite Braak et al. (2011) for the point about tau pathology being found earlier in the LC than in other brain regions.

There are unusual uses of capitalization throughout the text (why are Education, Locus Coeruleus, Noradrenaline, Cognitive Reserve, etc. capitalized?)

Throughout text- “beta pathology” is used, which is not an appropriate term. Should be specified as “beta-amyloid” or another more specific term

Abstract: starts off saying “to test this theory” without specifying what theory

Abstract: “…as well as biological brain maintenance the three groups” missing word ‘between’?

Page 2, line 43: Yakoov -> Yaakov

p. 3: It will likely be unclear to readers what “baseline functioning” is in the description of the Wilson et al. study. 

p. 3 line 92: LC abbreviation is defined but then not used or used inconsistently throughout the paper

p. 3 line 93: “irroration”? what are the authors referring to?

In regards to the relationship between LC structure and cortical grey matter, there is a relevant previous paper that found positive relationships between LC MRI contrast and structural thickness in various cortical regions (in particular frontoparietal regions) (Bachman, Dahl, Werkle-Bergner et al. 2021).

p. 4 line 145: citation for TMT-A underlying cognitive abilities?

p. 7 line 265- smoothing should be specifically stated as a 2 mm^3 FWHM kernel as opposed to the (2 2 2) notation used by the authors, which may not be understandable for readers

Figure 3: the results are thresholded at p<0.001 for AD and MCI and at p<0.001 FWE for HC. Why the different thresholds?

p. 12: What do c_ps and c_cs refer to? And “T”? 

p. 12: Why is it stated that the total effect of X on Y in HC was not significant when the confidence interval range did not include 0? What are the confidence intervals being compared with? Likewise, why is the direct effect of X on Y stated as being significant when the confidence interval range for that test did include 0?

p. 12 line 439 references the wrong figure

p. 16: ‘Histopathological evidence shows that in the early stages of neurodegeneration, the metencephalic areas appear to be less affected than cortical areas.” Some of the references cited actually make the opposite case, that brainstem nuclei are affected in the earliest pre-clinical stages of AD.

Author Response

Dear Reviewer 2, please find attached the comment and revisione made following your suggestions.

Round 2

Author Response

In the attached document we provided responses
